# CONTROLLABLE TEXT-TO-SPEECH SYNTHESIS WITH MASKED-AUTOENCODED STYLE REPRESENTATION

## ABSTRACT

Controllable text-to-speech (TTS) systems aim to manipulate various stylistic attributes of generated speech. Despite considerable research in this area, existing models that use natural language prompts as an interface often lack the ability for fine-grained control and face a scarcity of high-quality data. To address these challenges, we propose a two-stage style-controllable TTS system with language models, utilizing a masked-autoencoded style representation as an intermediary. In our approach, we employ a masked autoencoder to learn a content-disentangled style feature of speech, which is then discretized using a residual vector quantizer. In the first stage, an autoregressive transformer is used for the conditional generation of these style tokens from text and control signals. In the second stage, we generate codec tokens from both text and sampled style tokens. Experiments demonstrate that training the first-stage model on extensive datasets enhances the robustness of the two-stage model in terms of quality and content accuracy. Additionally, our model achieves superior control over attributes such as pitch and emotion. By selectively combining discrete labels and speaker embeddings, we can fully control the speaker's timbre and other stylistic information, or adjust attributes like pitch and emotion for a specified speaker. Audio samples are available at `https://style-ar-tts.github.io`.

## 1 INTRODUCTION

Controllable text-to-speech (TTS) systems aim to generate high-fidelity speech while allowing control over various style attributes of the synthesized speech, such as speaker timbre, pitch level and variation, emotion, acoustic environment, etc. Due to its promising applications in digital media production and human-computer interaction, controllable TTS has been attracting growing interest in the machine learning community with a substantial amount of research working on it (Guo et al., 2023; Leng et al., 2023; Ji et al., 2024; Yang et al., 2024; Zhou et al., 2024).

Despite the extensive research on this topic, controllable TTS still faces some unsolved challenges: **(1) Control Interface Issue**. Most existing works use natural language prompts as a medium of style control, which is friendly for non-professional users. However, style descriptions with natural language tend to be broad and coarse-grained, making it difficult to precisely control specific attributes. Moreover, the rich diversity of natural language brings more challenges to modeling the relationship between style attributes and prompts. It is also difficult to fully encompass the user instructions in real-world scenarios, restricting the application of these methods. **(2) Data Issue**. The training of well-performed TTS systems relies on high-quality speech corpora, which are often limited in both data volume and stylistic diversity. When using natural language as the control interface, the additional cost of generating prompt sentences further restricts the data size. Present controllable TTS datasets (Guo et al., 2023; Ji et al., 2024) are often limited to hundreds of hours. This constraint puts challenges on learning precise control abilities and improving generation diversity.

In this paper, we propose a fine-grained controllable TTS system. In contrast to natural language prompts, We divide the value ranges of various stylistic attributes of speech into multiple intervals, each represented by a label, and use these labels as conditional inputs to achieve fine-grained control. By selectively combining these labels with speaker embeddings, we can generate new speaker timbre while controlling other attributes, or adjust certain attributes such as emotion for a given speaker.

Our controllable TTS system adopts a two-stage generation paradigm using two language models (LMs), with a style representation as an intermediate output. We adopt a masked autoencoder (MAE) which learns to capture content-disentangled style information by reconstructing mel filterbank from the encoded content input and masked fbank. The features extracted by the style encoder of the trained MAE are then discretized and used as an intermediary of the TTS pipeline. Each of the two stages relies on a decoder-only transformer. The first stage generates style tokens conditioned on content and control signals, while the second stage generates codec tokens from the content input and the predicted style tokens. Due to low dependence on high-quality corpora, the style token generation phase can scale up to a large amount of data, boosting control capability and generation diversity; while in the codec token generation stage, a relatively small amount of data is sufficient to learn how to reconstruct codec tokens from content and style units, addressing the issue of high-quality data scarcity. To enhance the control accuracy of fine-grained attributes, we introduce classifier-free guidance to the style token generation stage. Moreover, considering the correlations between control signals, we discuss methods to determine the range of low-level label intervals from high-level labels to reduce information conflicts. Experiments indicate that our model achieves good style control ability while keeping decent audio quality and content accuracy.

## 2 RELATED WORKS

### 2.1 CONTROLLABLE TEXT-TO-SPEECH

Controllable TTS aims to enable control over stylistic attributes of the speech during synthesis. The earliest exploration, PromptTTS (Guo et al., 2023), extracts textual features from prompts with a fine-tuned BERT and incorporates them in a TTS backbone with attention. InstructTTS (Yang et al., 2024) achieves a text-controlled expressive TTS system with cross-modal representation learning. PromptTTS 2 (Leng et al., 2023) employs a variational network to generate reference acoustic features conditioned on text features. Audiobox (Vyas et al., 2023) builds a unified natural-language-instructed flow-matching model integrating speech, music, and audio generation. TextrolSpeech (Ji et al., 2024) integrates natural language style prompt into the condition of VALL-E (Wang et al., 2023a) for controllable TTS. VoxInstruct (Zhou et al., 2024) merges the content input and style prompt of TTS into a single composite textual instruction and utilizes a multimodal codec language model as the backbone for TTS. Unlike these methods using natural language as the control interface, we adopt a two-stage controllable TTS system with attribute labels for fine-grained control.

### 2.2 SPEECH ATTRIBUTE DECOUPLING

Some works attempt to obtain speech representations that disentangle content and stylistic attributes through self-supervised learning or other methods, facilitating applications like voice conversion, acoustic attribute editing, and controllable TTS. NANSY (Choi et al., 2021) deconstructs input speech into multiple information flows explicitly, and then reconstructs speech from these flows, obtaining a model capable of voice conversion, pitch shift, and other applications. DSVAE Lian et al. (2022b;a; 2023) presented a self-supervised method to disentangle content information and global speaker information, in an end-to-end manner. Prosody-TTS (Huang et al., 2023) utilizes an MAE to learn a prosody representation disentangled from content and speaker timbre, boosting expressive TTS. SpeechTokenizer (Zhang et al., 2023) decouples different aspects of speech information hierarchically across different quantizer layers by distillation, thereby unifying semantic and acoustic representations; NaturalSpeech 3 (Ju et al., 2024) proposes a codec that factorizes speech into individual subspaces representing different attributes like content, prosody, timbre, and acoustic details, facilitating modeling of intricate speech. In this paper, we adopt a masked autoencoder to extract content-disentangled style features of the speech, which are then used as an intermediate representation to facilitate controllable TTS.

## 3 METHOD

### 3.1 OVERVIEW

Our controllable TTS system consists of two major stages with a discrete style token as an intermediate representation. This style representation is from a transformer-based MAE as illustrated in

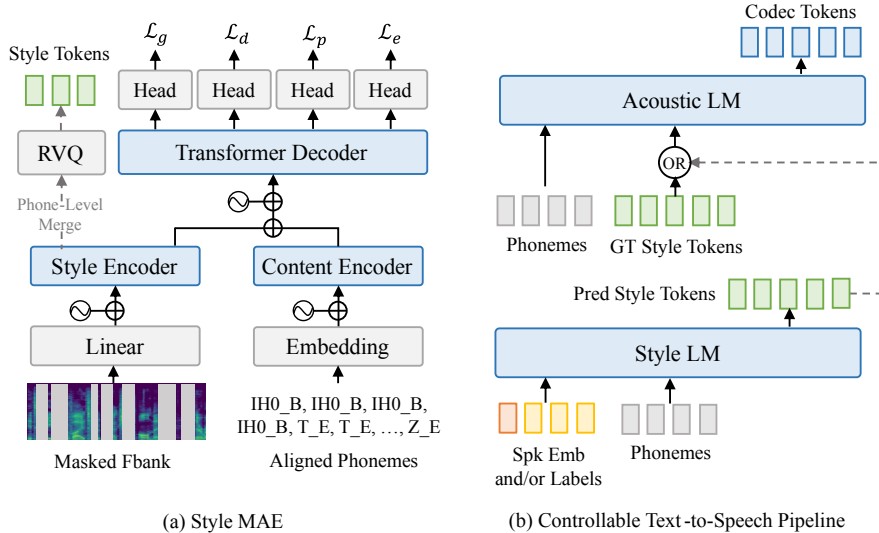

Figure 1: Model overview of our controllable TTS system. Figure (a) shows the architecture of the style MAE. Figure (b) illustrates the two-stage controllable TTS pipeline. The gray dashed lines represent paths that occur only during inference.

figure 1 (a), which learns to capture style information including speaker timbre, prosody, and acoustic environment in the speech with a mask-reconstruction paradigm. The style tokens of a speech clip can be extracted with the style encoder of the pre-trained MAE followed by a residual vector quantizer (RVQ) trained individually. The two stages of TTS are (1) **style token (ST) generation**, which generates style tokens conditioned on content phonemes and style controlling signals including discrete labels and / or continuous speaker embeddings; and (2) **codec token (CT) generation**, which generates codec tokens conditioned on content phonemes and style tokens, where the style tokens are either extracted from ground truth speech or predicted by the former stage. The generated codec tokens are then used to reconstruct the waveform with the codec decoder. Each of the two stages relies on a decoder-only transformer to conduct LM-style generation, as illustrated in figure 1 (b). We provide details of these modules respectively in the following subsections. Details of model configurations are provided in appendix A.

## 3.2 STYLE MASKED AUTOENCODER AND STYLE TOKENIZATION

The style masked autoencoder aims to learn to extract style information other than content, like speaker timbre, prosody, and acoustic environment by reconstructing mel filterbanks from masked ones and an additional content input with reconstruction and several auxiliary losses. Its architecture is illustrated in figure 1 (a). The two branches of input, which are masked fbanks and a temporal-aligned phoneme sequence where each phoneme is duplicated by its duration, are processed by two encoders separately. Both the style encoder and content encoder are multi-layer transformer encoders. The output of the two encoders together with sinusoidal positional embedding are added and fed to the transformer decoder.

Following Huang et al. (2023), we append four different linear heads at the end of the decoder for output projection used for different optimization objectives. The four objectives are (1) reconstruction loss $\mathcal{L}_r$: mean square loss between the masked fbank patches and the output of the reconstruction head; (2) contrastive loss $\mathcal{L}_c$: InfoNCE loss to maximize the similarity between the head output and the corresponding fbank patch, while minimizing its similarity with non-corresponding fbank patches; (3) pitch classification loss $\mathcal{L}_p$ and (4) energy classification loss $\mathcal{L}_e$, which are cross-entropy losses calculated on log-scale fundamental frequency (f0) and the L2-norm of the amplitude spectrogram from short-time Fourier transform, respectively, both of which are frame-level and binned to 256 scales. The final loss is a linear combination of the four losses:

$$\mathcal{L} = \lambda_r \mathcal{L}_r + \lambda_c \mathcal{L}_c + \lambda_p \mathcal{L}_p + \lambda_e \mathcal{L}_e \tag{1}$$

where $\lambda_r = 10$, and $\lambda_c$, $\lambda_p$, $\lambda_e$ are all 1. Intuitively, this design enables the MAE to extract content information from the encoded feature of the aligned phonemes, while extracting style information from the encoded feature of the masked fbank for reconstruction. Once the MAE finishes training, its style encoder should be capable of extracting style information from speech.

To reduce the sequence length for language modeling and eliminate redundant information in the style features, we conduct phone-level merge by averaging the frame-level features in the range of each phoneme. After that, we train an RVQ with 3 codebooks over the phone-level style features for discretizing the style representation for LM-style modeling. The RVQ module is trained independently from the MAE. Subsequent codec generation results using phonemes and ground truth style tokens demonstrate that this style representation encapsulates rich style information with little content leakage.

### 3.3 TWO-STAGE LM-STYLE CONTROLLABLE TEXT-TO-SPEECH

We use a decoder-only transformer for autoregressive generation for both of the two stages. Specifically, we adopt the multi-scale transformer as the backbone model (Yang et al.; Huang et al., 2024), which utilizes a stacked global-local transformer architecture to handle detailed acoustic token modeling and has exhibited remarkable capabilities in audio synthesis and modeling intrinsic relationships among different modalities, while remaining high efficiency in generating long sequences based on sub-quadratic self-attention. During training, the conditional inputs and target outputs are concatenated into a single sequence and fed to the transformer, with each part having a modality-specific *start* and *end* token at both ends. The LMs model the conditional distribution using next-token prediction with cross-entropy loss calculated on the target output part.

**ST Generation** In the first stage, we adopt a style LM to generate style tokens from phonemes and control signals. This procedure can be formulated as:

$$\mathrm{P}(\mathbf{s}) = \prod_{t=1}^{T} \prod_{i=1}^{N} \mathrm{P}(s_t^i | \tau, c, \mathbf{s}_{<t}, \mathbf{s}_t^{<i}; \theta_s) \tag{2}$$

where $\mathbf{s}$, $\tau$, $c$, and $\theta_s$ are style tokens, phonemes, control signals, and model parameters, respectively. Here, the control signals can be a speaker embedding and / or discrete control labels. For discrete control labels, we include *age*, *gender*, *pitch mean* for average pitch, *pitch std* for the extent of pitch variation, emotion represented by *arousal*, *dominance*, and *valence*, *snr* for signal-noise rate, and *C50* for reverberation level. These labels are denoted by extracting attribute values with some tools and binning them to different levels. We can use all these labels to generate speech with a new speaker, or combine pitch and emotion labels with a speaker embedding to adjust these attributes on the basis of a reference speaker. The training data of this stage can be scaled up to large corpora to achieve higher style diversity and control accuracy.

**CT Generation** In the second stage, we adopt an acoustic LM to generate codec tokens from phonemes and style tokens. No additional control signal is used in this stage, as we assume that the style information is carried by the style tokens. During training, the model takes ground truth style tokens and learns to reconstruct codec tokens of the speech. In inference, the style tokens can be either ground truth ones for speech reconstruction, or predicted ones from the former stage for controllable TTS. This procedure can be formulated as:

$$\mathrm{P}(\mathbf{a}) = \prod_{t=1}^{T} \prod_{i=1}^{N} \mathrm{P}(a_t^i | \tau, \mathbf{s}, \mathbf{a}_{<t}, \mathbf{a}_t^{<i}; \theta_a). \tag{3}$$

where $\mathbf{a}$, $\tau$, $\mathbf{s}$, and $\theta_a$ are codec tokens, phonemes, style tokens, and model parameters, respectively. We observe in our experiment that several hundred hours of data are sufficient for the model to learn to reconstruct speech of decent quality from phoneme and style tokens, therefore addressing the scarcity issue of high-quality corpora for controllable TTS.

### 3.4 CLASSIFIER-FREE GUIDANCE

We observe that for attributes with distinct differences among categories (like gender), simply adding the label to the prefix condition sequence leads to pretty good control capability. However, for attributes with fine-grained levels and relatively ambiguous boundaries, this simple approach fails to

Table 1: Training and inference datasets for different stages.

| Stage | Training Datasets | Inference Samples |
|---|---|---|
| Style MAE | Gigaspeech-xl, Librispeech | |
| Style LM | Gigaspeech-xl | small sets from |
| Acoustic LM | LibriTTS | LibriTTS (184), Gigaspeech (173), DailyTalk (201) |

Table 2: Comparing reconstructed speech from phonemes and ground truth style tokens to original speech, compressed speech and zero-shot TTS results.

| Method | LibriTTS | | | Gigaspeech | | |
|---|---|---|---|---|---|---|
| | SIM | MCD | UTMOS | SIM | MCD | UTMOS |
| GT. | / | / | $4.06 \pm 0.05$ | / | / | $3.47 \pm 0.10$ |
| GT. + Codec | 0.94 | 1.98 | $3.43 \pm 0.06$ | 0.91 | 2.21 | $2.87 \pm 0.09$ |
| YourTTS | 0.91 | 6.12 | $3.61 \pm 0.09$ | 0.85 | 6.72 | $2.33 \pm 0.09$ |
| XTTS-V2 | 0.91 | 5.96 | $3.68 \pm 0.08$ | 0.87 | 6.48 | $3.26 \pm 0.10$ |
| Acoustic LM + GT Style | 0.90 | 3.19 | $3.63 \pm 0.05$ | 0.86 | 3.68 | $3.24 \pm 0.08$ |

achieve satisfactory control accuracy. To enhance the model's control capabilities, we introduce classifier-free guidance (CFG) (Ho & Salimans, 2021), which is initially used in score-based generative models and performs well in aligning conditional input and results. We investigate CFG in the ST generation stage.

Specifically, during the training of the style LM, we randomly replace the controlling labels with a special empty control token with a probability of $p = 0.15$. During inference, for each position $i$, we apply correction to the logit value of style token $s_i$ with the formula

$$\log \hat{P}(s_t^i | \mathbf{s}_{<t}, \mathbf{s}_t^{<i}, \tau, c; \theta_s) = \gamma \log P(s_t^i | \mathbf{s}_{<t}, \mathbf{s}_t^{<i}, \tau, c; \theta_s) + (1-\gamma) \log P(s_t^i | \mathbf{s}_{<t}, \mathbf{s}_t^{<i}, \tau, \varnothing; \theta_s) \quad (4)$$

where $\tau$, $c$, and $\gamma$ represent text (phonemes), control labels, and guidance scale, respectively. The recalculated logit is then used for calculating the probability for sampling with the softmax function. This technique significantly improves the style coherence between the generated speech and the fine-grained control labels, boosting the control capability of the model.

We observe that when conducting random drop and logit re-calculating on speaker embeddings, the synthesized speech tends to have suboptimal quality with artifacts and pronounced background noise. Therefore, we only conduct CFG over discrete control labels.

## 4 EXPERIMENTS

### 4.1 DATASET AND STYLE ATTRIBUTES LABELING

The datasets used for different stages are illustrated in table 1. We adopt large-scale corpora for training the style MAE, where we combine Gigaspeech-xl (Chen et al., 2021) and Librispeech (Panayotov et al., 2015). We use Gigaspeech-xl solely for training the style LM, and use high-quality LibriTTS (Zen et al., 2019) with a relatively small scale for training the acoustic LM. For evaluation, we randomly pick approximately 200 samples respectively from LibriTTS, GigaSpeech, and a dialogue dataset, DailyTalk (Lee et al., 2023), to evaluate the models' performance across different data domains. We note the specific sizes of each test set after the dataset it is sampled from in the table.

To train the style LM, we need to label the different attributes of the data. We utilize multiple annotation tools to extract continuous values or classification probabilities for different speech attributes, and split them into different bins by performing equidistant division within an upper and lower boundary that covers most of the data to obtain the discrete control labels. Details of labeling tools and splitting strategies are provided in appendix B.

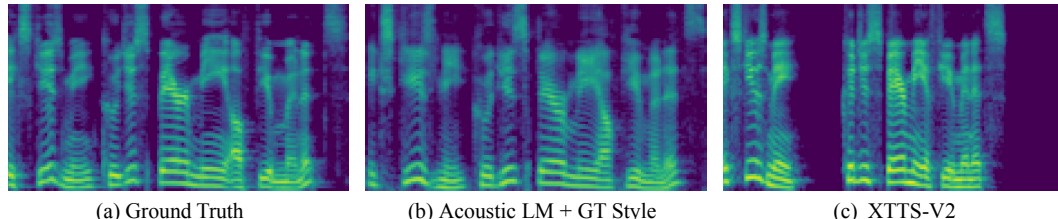

| (a) Ground Truth | (b) Acoustic LM + GT Style | (c) XTTS-V2 |

Figure 2: Spectrogram from original speech, reconstructed speech with ground truth style tokens and zero-shot TTS result.

## 4.2 METRICS

Our evaluation of model performance primarily consists of speech naturalness, content accuracy, speaker similarity, speech reconstruction quality, and control accuracy. We use objective metrics for evaluation. For speech naturalness, we adopt UTMOS (Saeki et al., 2022) to predict the MOS score of each sample and report mean values and 95% confidence intervals for each test set. For content accuracy, we use Whisper large-v3 (Radford et al., 2022) to transcribe the speech and calculate the word error rate (WER) against the ground truth text. For speaker similarity, we compute cosine similarity on speaker embedding extracted by wavlm-base-plus-sv [1]. For reconstruction quality, we calculate MCD between generated and ground truth speech with tools provided in fairseq [2]. For control accuracy, we use the annotation tools to extract attribute labels and compute percentage accuracy with ground truth labels. Considering the challenges of achieving precise control with fine-grained labels, we make some relaxation that results differing from the ground truth attribute label by one bin are also considered correct.

## 4.3 RESULTS AND ANALYSIS

### 4.3.1 RECONSTRUCT SPEECH STYLE FROM STYLE TOKENS

To validate that our style tokens encapsulate rich voice style information, we reconstruct speech from phonemes and ground truth (GT) style tokens, and compare them with original speech, compressed speech from the codec, and zero-shot TTS results. We use YourTTS (Casanova et al., 2022) and XTTS-V2 (Casanova et al., 2024) as representative zero-shot TTS systems for comparison. The results on LibriTTS and Gigaspeech are shown in table 2. For results on both test sets, our model achieves comparable UTMOS to recent zero-shot TTS systems. This demonstrates the reliability of our model in terms of speech naturalness. Besides, our model achieves comparable speaker similarity with zero-shot TTS systems, indicating that the style tokens contain rich speaker information for speech synthesis. Moreover, the reconstruction results have significantly lower MCD than zero-shot TTS, proving that it is closer to the original audio in terms of prosody and other style information like acoustic environment, which further validates the effectiveness of our style MAE.

Figure 2 illustrates the spectrogram of some sample results on DailyTalk. It can be observed that despite some over-smoothing in certain details, the acoustic LM is able to leverage the style information contained in the style tokens to achieve accurate reconstruction on out-of-domain samples, indicating the effectiveness of our style representation. In contrast, zero-shot TTS that only leverages speaker information cannot achieve prosody reconstruction. On the other hand, when we swap the phonemes and style tokens from different samples, the model fails to generate meaningful speech. This demonstrates that our style representation does not result in significant content information leakage, and contains more complex stylistic information beyond simple speaker, pitch, and energy attributes.

### 4.3.2 CONTROLLABLE TTS WITH DISCRETE LABELS

---

[1] https://huggingface.co/microsoft/wavlm-base-plus-sv

[2] https://github.com/facebookresearch/fairseq/blob/main/examples/speech_synthesis/docs/ljspeech_example.md#mcdmsd-metric

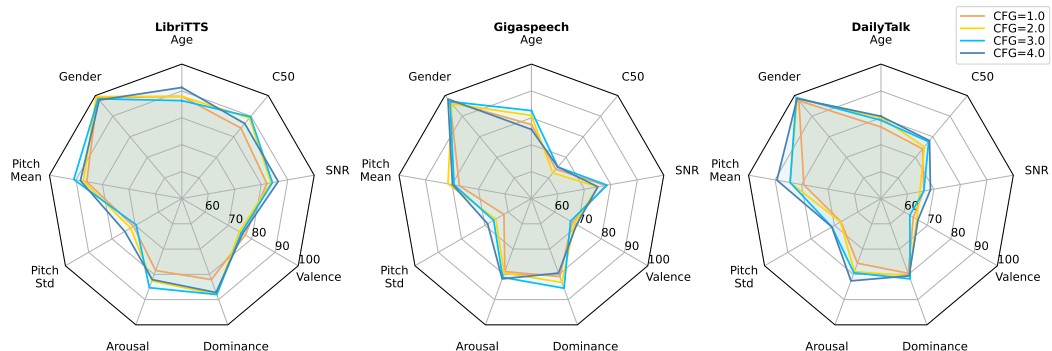

Figure 4: Control accuracy of the two-stage controllable TTS with discrete labels under different CFG scales. The coordinate range is also set to 50-100.

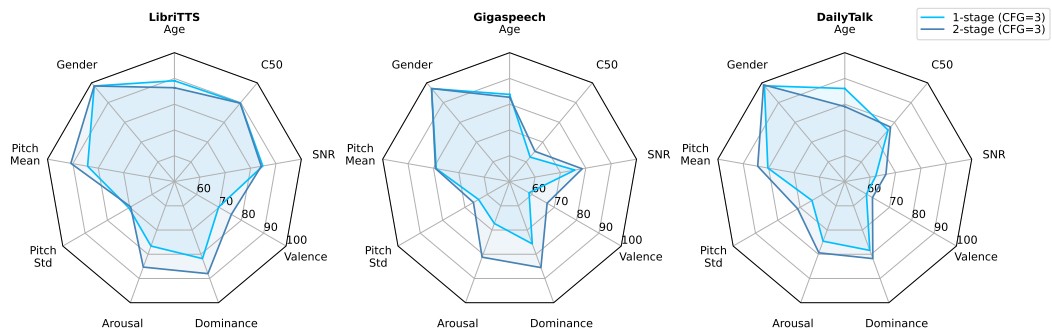

Figure 5: Control accuracy of the one-stage and two-stage controllable TTS with discrete labels under a CFG scale of 3.0. The coordinate range is set to 50-100 for the more apparent differences.

In this section, we evaluate the performance of our controllable TTS system with solely discrete labels. To validate the effectiveness of our two-stage design, we train a one-stage model as the baseline, which generates codec tokens from phonemes and control labels directly. We use LibriTTS for training the one-stage model, which is the same as training the acoustic LM. Due to

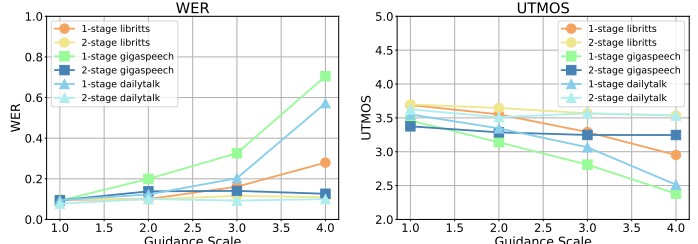

Figure 3: WER and UTMOS on different guidance scales.

the sheer magnitude of their quantity, traversing all possible attribute combinations is not feasible. Furthermore, the correlation among attributes may render certain combinations of labels impossible or difficult to achieve. Therefore, we use label combinations extracted from ground truth speech for control and evaluation and further modify specific attributes for case studies.

We first consider the content accuracy and naturalness of the TTS systems. We illustrate the WER and UTMOS values of the two models under different CFG scales in figure 3. It can be seen that for the one-stage model trained on LibriTTS, as the CFG scale increases, the word error rate rises and UTMOS declines, especially on out-of-domain test sets of Gigaspeech and DailyTalk, manifesting significant degradation in content accuracy and naturalness. This indicates the instability of the one-stage model trained on small, high-quality datasets when subjected to an increased CFG scale, making it difficult to balance control capabilities with speech quality. On the other hand, the two-stage model with the first stage trained on large corpora exhibits good and stable content accuracy and naturalness with growing CFG scales. This proves that the first stage trained on extensive data helps in enhancing the content robustness of controllable TTS, without affecting speech quality by error propagation.

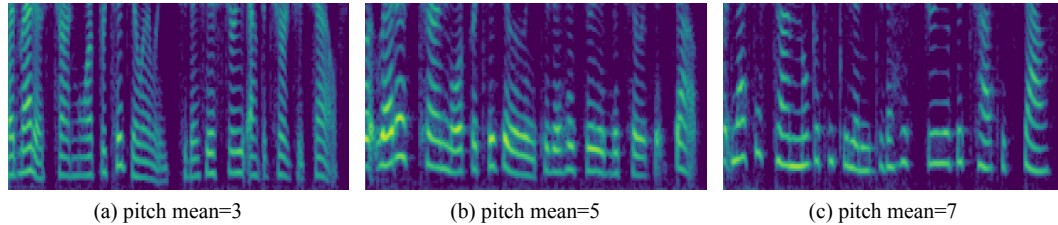

|  (a) pitch mean=3 | (b) pitch mean=5 | (c) pitch mean=7 |

Figure 6: Spectrograms obtained using pitch labels of different levels in two-stage controllable TTS.

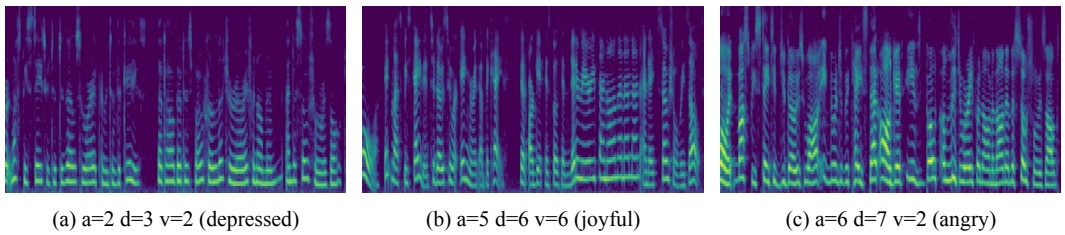

|  (a) a=2 d=3 v=2 (depressed) | (b) a=5 d=6 v=6 (joyful) | (c) a=6 d=7 v=2 (angry) |

Figure 7: Spectrograms obtained using different compositions of emotion labels in two-stage controllable TTS.

In figure 4, we illustrate the control accuracies of the two-stage model under different CFG scales. We can see that the effect of CFG varies for different attributes. For gender attributes with fewer categories and significant differentiation, the presence or absence of CFG shows no clear impact and the model achieves good control performance in both cases. However, for fine-grained attributes like *age* and *pitch std*, appropriate CFG scales can benefit control accuracy significantly. This indicates that CFG plays a crucial role in the precise control of fine-grained attributes. Meanwhile, we find that larger CFG scales are not always beneficial. For some attributes, control accuracy initially increases before subsequently declining as the scale rises. We speculate that this may be due to larger scale values causing distortion in the generated speech, similar to the phenomenon observed with CFG in score-based models. The full results of these two models under different CFG scales are provided in appendix C.

We further evaluate the control ability of the models. In figure 5, we compare the control accuracies of the one-stage and the two-stage model under a CFG scale of 3.0. It can be seen that the two-stage model demonstrates comparable control capabilities over timbre attributes like *age* and *gender*, while achieving superior control over emotion attributes across all three test sets. For controlling average pitch, the two-stage model performs better on LibriTTS and DailyTalk. And it also achieves better accuracy over pitch variation, SNR, and C50 on Gigaspeech and DailyTalk. This indicates that compared to the one-stage model trained on high-quality corpora with limited scale, the two-stage model with the first stage trained with extensive data boosts modeling diverse pitch and acoustic conditions. The limitations of DailyTalk (e.g., only containing two speakers) indeed negatively impact the model's performance in age control for the two-stage model.

Taking *pitch mean* and emotion labels as examples, we plot the spectrograms to illustrate the effects of modifying specific attributes of the given samples. Figure 6 showcases the results using different average pitch labels while keeping the content and other attributes constant. We only display the frequency range of 0-2kHz for clearer visualization. It can be seen that when we raise the value of the *pitch mean* label, the fundamental frequency levels up, and the distance between formants increases, indicating that the speaker timbre grows shriller, proving the effectiveness of our model on controlling average pitch. In figure 7, we use three different groups of emotion labels for one test sample. The spectrogram shows that labels corresponding to elevated emotion lead to more pronounced pitch variation compared to those of subdued emotion. We refer the reader to our demo page for more samples.

Table 3: Results of controllable TTS combining speaker embedding, pitch and emotion labels.

| Test set | Model | CFG Scale | WER(%) | SIM | P.S. | Aro. | Dom. | Val. | UTMOS |
|----------|-------|-----------|--------|-----|------|------|------|------|-------|
| Gigaspeech | 1-stage | 1.0 | 12.4 | 0.86 | 57.5 | 68.8 | 75.1 | 65.3 | 3.45 ± 0.07 |
| | | 2.0 | 11.9 | 0.86 | 59.2 | 70.8 | 76.3 | 68.2 | 3.41 ± 0.07 |
| | | 3.0 | 13.5 | 0.86 | 62.7 | 74.0 | 78.6 | 65.0 | 3.37 ± 0.07 |
| | | 4.0 | 14.8 | 0.85 | 64.2 | 71.7 | 75.4 | 64.2 | 3.30 ± 0.07 |
| | 2-stage | 1.0 | 13.3 | 0.85 | 61.3 | 76.3 | 76.9 | 65.3 | 3.33 ± 0.08 |
| | | 2.0 | 13.3 | 0.84 | 66.5 | 79.8 | 81.2 | 67.6 | 3.23 ± 0.09 |
| | | 3.0 | 14.0 | 0.85 | 69.4 | 76.9 | 79.8 | 65.9 | 3.26 ± 0.09 |
| | | 4.0 | 14.9 | 0.86 | 68.5 | 80.9 | 82.7 | 68.2 | 3.27 ± 0.09 |
| DailyTalk | 1-stage | 1.0 | 12.1 | 0.80 | 51.7 | 67.9 | 75.4 | 57.7 | 3.36 ± 0.06 |
| | | 2.0 | 12.8 | 0.81 | 60.9 | 67.9 | 77.9 | 58.5 | 3.27 ± 0.07 |
| | | 3.0 | 16.1 | 0.80 | 65.7 | 73.4 | 79.1 | 59.7 | 3.23 ± 0.07 |
| | | 4.0 | 16.3 | 0.80 | 64.4 | 72.4 | 80.6 | 62.7 | 3.16 ± 0.07 |
| | 2-stage | 1.0 | 12.0 | 0.80 | 62.4 | 72.9 | 76.4 | 59.5 | 3.35 ± 0.09 |
| | | 2.0 | 10.7 | 0.79 | 68.9 | 77.6 | 81.1 | 61.9 | 3.39 ± 0.08 |
| | | 3.0 | 11.1 | 0.80 | 73.1 | 75.9 | 77.4 | 60.2 | 3.37 ± 0.08 |
| | | 4.0 | 12.9 | 0.80 | 74.4 | 77.9 | 77.6 | 66.9 | 3.37 ± 0.08 |

Table 4: Pearson correlation coefficients between high-level and low-level attributes.

| Low-Level \ High-Level | Age | Gender | Arousal | Dominance | Valence |
|------------------------|-----|--------|---------|-----------|---------|
| Pitch Mean | -0.15 | -0.74 | 0.38 | 0.29 | 0.06 |
| Pitch Std | -0.01 | 0.37 | 0.39 | 0.33 | 0.06 |

### 4.3.3 CONTROLLING PITCH AND EMOTION WITH A REFERENCE SPEAKER

In this section, we present the results that alternate the timbre-related and acoustic condition labels including *age*, *gender*, *SNR* and *C50* with speaker embedding from WeSpeaker (Wang et al., 2023b) to achieve control over pitch variation and emotion with a specified reference speaker. The *pitch mean* label is kept in the condition but we don't use it for control. We present results on Gigaspeech and DailyTalk in table 3. It can be seen that our model achieves decent speaker similarity on both test sets as well as comparable control accuracy to the discrete-label-only paradigm over pitch variation and emotion. This indicates the effectiveness of our model on control attributes like pitch and emotion for a specified speaker. Moreover, compared to fully-discrete-label controlling, the one-stage model shows better content robustness with growing CFG scale in this setting, and the one-stage and two-stage models exhibit comparable performance in content accuracy and speaker similarity. Despite this, the two-stage model retains advantages in control over pitch variation, arousal, and dominance, demonstrating that the ST generation model trained on an extensive dataset remains advantageous in modeling pitch-related stylistic information in this setting.

### 4.4 CORRELATION AMONG CONTROL ATTRIBUTES

In fact, the information contained among different attributes may overlap, manifesting as correlations between labels. Certain high-level attributes can be reflected in lower-level acoustic properties. For example, attributes related to speaker timbre, such as age and gender, are closely linked to average pitch, while emotion is closely related to pitch variation. In table 4, we present the Pearson correlation coefficients (Wikipedia, 2024) between high-level attributes and pitch attributes calculated on LibriTTS. It can be seen that age is correlated with average pitch to some degree, while gender, arousal, and dominance show significant correlations with both the mean and variation of pitch, indicating the presence of overlapping information. Additionally, the limited performance of the annotation tools may also leads to significant correlation among different emotional dimensions. Theoretically, the three dimensions of arousal, dominance, and valence are orthogonal. However,

as shown in figure 8, the distributions of arousal and dominance extracted by the model exhibit a strong linear correlation.

Due to the correlation among different attributes, using control signals that contain conflict information may lead to sub-optimal speech quality and control capability. We showcase examples on our demo page where conflicting control signals lead to degraded control performance. To achieve better control accuracy and content quality, we can restrict the ranges of low-level attributes with desired high-level attribute labels, thereby avoiding information conflicts. A straightforward solution is a statistical approach, where we can calculate the conditional distributions of *pitch mean* and *pitch std* given other labels on the training dataset, and sample labels from the distribution. Another solution is a learning-based method, where we can train label predictors for estimating low-level attributes from the given high-level labels. We train two 3-layer MLPs with a hidden dimension of 160 to predict *pitch mean* and *pitch std* from *age*, *gender*, *arousal*, *dominance* and *valence*. We find that the accuracy of predicting *pitch mean* and *pitch std* can reach around 40%, while the soft accuracy—considering a label error of no more than 1 as correct—exceeds 80%. This

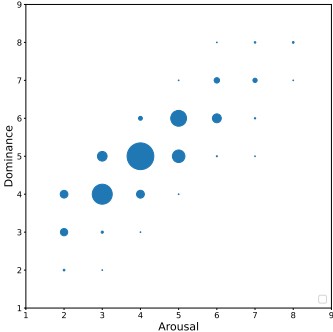

Figure 8: Illustration of the data distribution for arousal and dominance.

demonstrates the effectiveness of these predictive models. Once these models finish training, the output probabilities can be used to sample pitch labels.

## 5 CONCLUSION

In this paper, we propose an LM-based fine-grained controllable TTS system. We adopt a two-stage generation pipeline, with an autoregressive transformer as the backbone for each stage. We design a masked autoencoder for extracting content-disentangled style features from the speech and use the discretized feature as the intermediate output of the TTS pipeline. By selectively combining discrete control labels with speaker embeddings, our model supports both generating new speaker timbre while controlling other attributes, and controlling emotion and pitch variation for a specified speaker. Experiments indicate the effectiveness of our model.

In the future, we may explore more diverse control signals and employ techniques such as prompt engineering to integrate large language models with controllable TTS, enabling support for both natural language prompts and fine-grained control signals.

## 6 LIMITATIONS

Despite that our approach achieves fine-grained control over multiple style attributes, our method and evaluation protocols still suffer from several limitations: 1) Due to the performance limitations of labeling tools, there may be errors in the attribute annotations of the training data, which could lead to a decline in the model's control capabilities. 2) Evaluation with label combinations from real data may present issues of uneven distribution, particularly for attributes with significant distribution bias, such as SNR and C50. Therefore, the evaluation may not fully accurately reflect the model's control capabilities. 3) Due to their small proportion in the training data, some marginal label combinations may lead to degraded generated audio and diminished control performance. We will explore solutions to these issues in future work.

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

Table 5: Hyper-parameters of different modules of our approach.

| Model | Hyperparameter | |
|---|---|---|
| Style MAE | Encoder Layers | 12 |
| | Decoder Layers | 2 |
| | Hidden Dimension | 768 |
| | Mask Probability | 0.75 |
| | Fbank Channels | 128 |
| Style LM & Acoustic LM | Global Layers | 20 |
| | Local Layers | 6 |
| | Hidden Dim | 1,152 |
| | Global Attention Heads | 16 |
| | Local Attention Heads | 8 |
| | FFN Dim | 4,608 |

Table 6: Extracting tools and binning strategies for different attributes.

| Attribute | Extracting Tool | Lower Bound | Upper Bound | Bin Number |
|---|---|---|---|---|
| Age | w2v2-age-gender | 0.0 | 1.0 | 4 |
| Gender | w2v2-age-gender | 0 | 100 | 10 |
| Arousal, Dominance, Valence | w2v2-emotion | 0.2 | 0.8 | 7 |
| Pitch Mean | DataSpeech | 45.0 | 320.0 | 10 |
| Pitch Std | DataSpeech | 0.0 | 132.0 | 10 |
| SNR | DataSpeech | -9.16 | 77.13 | 10 |
| C50 | DataSpeech | 0.0 | 25.0 | 10 |

## A  IMPLEMENTATION DETAILS

In table 5, we illustrate the model hyper-parameters of the style MAE and two language models in our approach. For codec, we train a SoundStream (Zeghidour et al., 2021) model for 16k audio, with 8 quantization levels, a codebook size of 1024, and a downsampling rate of 320. We use the first 3 quantization levels only. We also use 3 RVQ layers for style tokens.

## B  STYLE ATTRIBUTE LABELING

In this section, we provide details of how we obtain the labels of different attributes. The extracting tools and binning strategies are summarized in table 6. For age and gender, we use a finetuned wav2vec2 model [3] to extract gender classification probability and estimated age between 0-100. We then split age into 4 categories: *male*, *neutral-masculine*, *neutral-feminine*, *female*, with the criteria being the probability of *male*, and thresholds of 0.65, 0.5 and 0.35.

For emotion labels, we adopt another finetuned wav2vec2 model [4] to extract the predicted logits of arousal, dominance, and valence. The range of the logits is 0-1, yet most audio falls between 0.2 and 0.8. Therefore, we divide the interval from 0.2 to 0.8 into seven labels with a distance of 0.1.

For pitch and acoustic conditions, we utilize DataSpeech (Lyth & King, 2024) to extract the mean value and standard variation of pitch, as well as SNR and C50. The ranges between the upper and lower bounds of each attribute are divided into 10 equidistant intervals, with the boundaries listed in the table.

## C  SUPPLEMENTARY EXPERIMENT RESULTS

In table 7, we provide the full results of the one-stage and two-stage models with discrete labels under different CFG scales, corresponding to figure 4 and figure 5 in section 4.3.2. This table provides a more accurate and comprehensive comparison of the performance between the one-stage and two-stage models, as well as the impact of CFG scales on both of them. It can be seen that CFG is effective in boosting control performance for both the one-stage and two-stage models. Moreover, the results demonstrate that the two-stage model outperforms the one-stage model in attributes such as pitch mean and arousal across a wide range of settings, further supporting the conclusions drawn in section 4.3.2.

---

[3] https://github.com/audeering/w2v2-age-gender-how-to
[4] https://github.com/audeering/w2v2-how-to

Table 7: Control accuracy of controllable TTS with discrete labels.

| Test set | Model | CFG Scale | Age | Gen. | P.M. | P.S. | Aro. | Dom. | Val. | SNR | C50 |
|---|---|---|---|---|---|---|---|---|---|---|---|
| LibriTTS | 1-stage | 1.0 | 69.6 | 96.7 | 76.1 | 63.6 | 74.5 | 78.0 | 73.6 | 73.4 | 89.7 |
| | | 2.0 | 88.6 | 98.9 | 86.4 | 66.6 | 81.8 | 82.9 | 70.7 | 86.4 | 90.2 |
| | | 3.0 | 89.1 | 98.4 | 84.2 | 70.4 | 76.6 | 81.8 | 69.8 | 84.8 | 89.7 |
| | | 4.0 | 93.5 | 97.8 | 89.1 | 71.5 | 67.1 | 77.4 | 63.9 | 75.0 | 88.6 |
| | 2-stage | 1.0 | 88.0 | 98.9 | 85.9 | 70.7 | 78.5 | 82.1 | 77.2 | 82.1 | 84.2 |
| | | 2.0 | 88.0 | 99.5 | 87.2 | 72.0 | 82.6 | 88.0 | 74.7 | 83.7 | 89.1 |
| | | 3.0 | 86.4 | 98.4 | 90.8 | 69.6 | 85.3 | 88.0 | 75.5 | 84.2 | 89.7 |
| | | 4.0 | 91.3 | 97.8 | 88.3 | 74.5 | 82.1 | 87.2 | 76.1 | 86.4 | 86.4 |
| Gigaspeech | 1-stage | 1.0 | 72.3 | 97.7 | 72.8 | 55.5 | 68.8 | 72.0 | 57.5 | 64.2 | 63.6 |
| | | 2.0 | 83.2 | 97.1 | 77.7 | 67.6 | 70.5 | 83.2 | 60.7 | 68.8 | 65.3 |
| | | 3.0 | 83.8 | 97.1 | 78.9 | 63.9 | 67.3 | 75.7 | 58.7 | 75.7 | 62.4 |
| | | 4.0 | 85.0 | 92.5 | 78.0 | 66.5 | 43.4 | 66.8 | 49.4 | 70.5 | 67.1 |
| | 2-stage | 1.0 | 77.5 | 97.1 | 77.2 | 61.8 | 78.9 | 80.9 | 67.6 | 78.0 | 64.2 |
| | | 2.0 | 80.9 | 96.0 | 81.5 | 65.3 | 79.5 | 83.2 | 68.5 | 75.1 | 62.4 |
| | | 3.0 | 82.7 | 97.1 | 79.2 | 66.2 | 81.2 | 85.5 | 66.8 | 78.6 | 65.3 |
| | | 4.0 | 75.7 | 98.3 | 79.8 | 68.8 | 81.8 | 79.5 | 69.4 | 75.1 | 65.3 |
| DailyTalk | 1-stage | 1.0 | 76.1 | 96.5 | 73.1 | 56.5 | 68.9 | 71.6 | 60.4 | 69.2 | 79.6 |
| | | 2.0 | 85.1 | 98.5 | 80.3 | 61.2 | 77.4 | 77.6 | 61.4 | 78.6 | 76.6 |
| | | 3.0 | 86.1 | 98.5 | 80.3 | 64.7 | 74.6 | 78.4 | 59.7 | 62.2 | 76.1 |
| | | 4.0 | 81.6 | 99.5 | 79.1 | 69.4 | 61.9 | 75.1 | 52.0 | 41.3 | 78.6 |
| | 2-stage | 1.0 | 76.6 | 97.5 | 79.1 | 67.2 | 75.6 | 79.6 | 63.9 | 66.2 | 74.1 |
| | | 2.0 | 80.1 | 98.5 | 84.3 | 67.4 | 78.9 | 80.6 | 65.4 | 64.7 | 75.1 |
| | | 3.0 | 79.1 | 99.0 | 84.3 | 71.1 | 79.4 | 81.8 | 62.4 | 66.2 | 77.6 |
| | | 4.0 | 80.6 | 98.5 | 89.3 | 71.1 | 82.6 | 80.6 | 65.9 | 68.7 | 78.1 |

