# OpenReview forum: "Controllable Text-to-Speech Synthesis with Masked-Autoencoded Style Representation"
_ICLR.cc/2025/Conference — ICLR 2025 Conference Withdrawn Submission_

### Official Review · Reviewer_S4Kw · 2024-10-15

**Soundness:** 2
**Presentation:** 2
**Contribution:** 3
**Rating:** 5
**Confidence:** 5

**Summary:**

This paper proposed a text-to-speech system that leverages masked autoencoder to extract discrete style representations of speech. The system follows a two-stage approach, where the style tokens are first generated and then the codec tokens, with two separate autoregressive transformers respectively. The authors claimed that this approach offers better controllability across speech attributes on the synthesis, and also enhances the quality and content fidelity of generated speech.

**Strengths:**

The approach of combining MAE and RVQ to extract quantized representations for non-linguistic speech information is novel, and can potentially inspire other similar fields, such as generative spoken language modeling. Additionally, the use of it on TTS to gain better controllability of speech styles is also original.

**Weaknesses:**

Clarity Issues:
 - For Equations (2) (3) (4), several notations are left unexplained. For example, what is $I$? What does $s_{<t}, s^{<i}_t$ mean? What is $\theta_s$. The authors provide these formulations but I don't feel like they are well defined. The authors should either remove those formulations or provide more explanation on them.
 - The methods in Table 2 are not well-explained. There is nowhere in the main text that explicitly explain what are "GT. + Codec" and "Acoustic LM + GT Style". While I can infer from the text that "Acoustic LM + GT Style" is passing the ground-truth style tokens and use the Acoustic LM to infer the codec tokens, I have no idea what is "GT. + Codec" though.
- Figure 3 is super-hard to read. It is too small. Please consider enlarge it.

I think most of my comments about the paper come from the experiments.
Experiments:
 - In Section 4.3.1, you are comparing existing TTS systems (YourTTS, XTTS-V2) that conditioned only on phonemes, with your system that conditioned on phonemes and ground truth style tokens. This is misleading as your system has an inherent advantage to access some ground truth attributes of the target speech. Furthermore, this is the only experiment you compare with the existing systems. My suggestion is to run experiments comparing your method with the whole two-stage inference and the existing TTS systems. Ideally, you should also compare with these methods trained with the similar datasets and number of parameters, where in this paper, the information is not given.
 - Also, in Section 4.3.1, you mentioned that "when we swap the phonemes and style tokens from different samples, the model fails to generate meaningful speech. This demonstrates that our style representation does not result in significant content information leakage." But to me, this sounds like a content leakage issue. As the syntheses of the swapped version should still be producing the same content as the given phoneme sequence.
 - You claimed that "training the first-stage model on extensive datasets enhances the robustness of the two-stage model in terms of quality and content accuracy". I don't think I saw experiments that support this. I thought that you will compare a model that the first-stage is trained on extensive datasets v.s. a ablated model that the first-stage is not trained on extensive datasets.
 - See more in the questions...

**Questions:**

- See weaknesses.
- Why does classifier-free guidance (CFG) provide you better controllability? It trains to interpolate between conditional and unconditional generation, but does it really allow you to "exaggerate the effect of the control signal"? The authors didn't do ablations to verify this. The authors should either conduct ablation study, or at least cite previous works that support this assumption.
- Is it fair to say two-stage approach is better than the single-stage approach? In Table 3, it seems like the 1-stage methods are consistently outperforming the 2-stage methods on GigaSpeech in terms of UTMOS and WER? The authors should explain this.
- The numbers in Table 3 and Figure 3 clearly don't match. For instance, there are >60% WER and <2.5 UTMOS in the Figure but not in the Table. The authors should make it clear what is happening.
- Why is the section called "reconstruct speech style from style tokens"? The Acoustic LM is generating the codecs from the ground truth style tokens. It's more like reconstructing the whole speech signal given the style tokens and phonemes?

---

### Official Review · Reviewer_DYKL · 2024-10-27

**Soundness:** 2
**Presentation:** 2
**Contribution:** 2
**Rating:** 3
**Confidence:** 4

**Summary:**

This paper introduces a two-stage TTS framework designed for controllable TTS generation by modifying style tokens. The first stage generates style tokens based on input phonemes and style control signals, while the second stage produces codec tokens using the generated style tokens and phoneme input. Additionally, the paper presents a novel classifier-free guidance (CFG) strategy to improve the style token generation process when using discrete control labels.

**Strengths:**

1. This paper addresses an important problem in TTS by enabling disentanglement and control over multiple style attributes.
2. The proposed classifier-free guidance approach for discrete control labels is both novel and practical.

**Weaknesses:**

1. The style tokens combine speaker timbre, prosody, and acoustic environment information into a single representation, primarily through a reconstruction task. However, the contributions of each attribute are not well disentangled. Additional experiments showing how modifying one attribute affects others would strengthen the analysis.
2. The experiments and demos focus heavily on the acoustic LM + ground truth (GT) style reconstruction results. However, the main contribution should emphasize style token generation rather than reconstruction.
3. The formatting and presentation of tables could be improved. For example, using only one CFG parameter for inference, except in the ablation study of CFG hyperparameters, would provide a clearer understanding for readers.

**Questions:**

In the demo, changing attributes other than timbre (such as reverberation or SNR) appears to affect the speaker's timbre. However, I assume that for these models,  the speaker embedding should be a fixed input. Could you clarify why the reference timbre is not preserved in these cases? Additionally, could you evaluate the similarity of other attributes when modifying only a single attribute?

---

### Official Review · Reviewer_g9hk · 2024-11-01

**Soundness:** 1
**Presentation:** 1
**Contribution:** 1
**Rating:** 1
**Confidence:** 5

**Summary:**

This paper studies the problem of controllable text to speech synthesis (TTS). It proposes an architecture where the input is comprised of a phoneme sequence representing the desired words to be spoken, a speaker embedding to control the speaker identity, and discrete control signals representing average pitch, emotion, age, gender, etc.

The proposed architecture first trains a frame-level speech style encoder ("Style MAE") trained via a masked reconstruction objective given masked spectrogram and time-aligned phonetic inputs. The outputs of this model are discretized with an RVQ stack and called "style tokens".

A second model (Style LM) is then trained to predict these style tokens given the target phoneme sequence and the control signals.

A third model (Acoustic LM) is then trained to take the predicted style tokens, along with the target phoneme sequence, and predict a set of neural codec tokens used to reconstruct the speech signal.

Experimental results are presented on speech reconstruction and controllable TTS.

**Strengths:**

The paper studies an interesting and timely problem, namely style controlled TTS.

**Weaknesses:**

The most egregious weakness of this paper is in the core experimental results on style-controllable TTS. Specifically:
- The paper does not compare against a single baseline model from the literature for this task, when numerous open source models already exist given the current popularity of the task. It only compares against a 1-stage version of the proposed model. This fact alone is sufficient to warrant a "Reject" for ICLR.
- Only automatic metrics are used, no human listening tests at all are conducted. Again, for a TTS paper this fact alone is sufficient to warrant a "Reject" for ICLR.

I also believe that the reconstruction results are flawed, specifically the comparison against voice cloning TTS models like YourTTS or XTTSv2. The proposed model is being given as input frame-level embeddings extracted from the speech signal used as a reconstruction target (in the form of the style tokens), whereas the voice cloning TTS baselines are not. Despite the authors' claims that the style tokens do not contain information about the phonetic content of the speech signal, I strongly believe they do (given that similar MLM encoders - without the extra phonetic input - such as HuBERT or WavLM represent the current SotA in the speech field for self-supervised learning of information like phonetic content). The authors would need to provide some experimental justification for their claim, but such evidence doesn't exist in the paper.

The model is very complicated, with 3 separately trained components (4 if you count the RVQ stack which is trained separately from the Style MAE. However, the paper does not conduct enough ablation studies to verify the importance of all of these components. For example, what happens if you simply replace the style tokens with off-the-shelf HuBERT units?

The paper is difficult to follow and missing critical details. For example:
- The paper does not describe which neural codec model is used, it only ever refers to the output of the Acoustic LM as "codec tokens"
- The paper does not describe how the discrete control signals are extracted during training, it only states "These labels are denoted by extracting attribute values with some tools and binning them to different levels."
- The training of the RVQ stack is never explained (which loss functions, codebook size, what specific type of RVQ, etc).

**Questions:**

See my numerous points in the "Weaknesses" section; given how severely lacking the experimental validation in this paper is, it would require numerous additional experiments and a complete re-write to merit acceptance to ICLR.

---

### Official Review · Reviewer_fjKH · 2024-11-04

**Soundness:** 3
**Presentation:** 4
**Contribution:** 3
**Rating:** 6
**Confidence:** 3

**Summary:**

This paper proposes an approach to train TTS systems controllable by a set of discrete labels (for age, pitch, emotion, etc.) via a two-stage process; the first stage converts the control labels to style tokens, while the second converts the style tokens to acoustic tokens. Here, the style tokens are learned via a masked autoencoder approach. The paper shows some interesting experiments, for e.g. how context-free guidance improves performance for certain label types and how training a two-stage model improves stability.

**Strengths:**

1. The paper is extremely well written and easy to understand.
2. The approach is elegant and wisely decomposes the task into two stages (style control labels to style tokens, and style tokens to acoustic tokens) that allow training each stage on differing amounts of data.
3. The experiments are interesting; I took away some good findings, such as the fact that CFG helps different types of conditioning labels differently and how different controls correlate.

**Weaknesses:**

1. The paper does not compare to existing baselines; it should do so, especially for easily available open-source ones that support some of the same controls used by this work. For example, Parler-TTS (https://github.com/huggingface/parler-tts) supports gender, pitch, snr and c50 control. Closed-source models (e.g. SpeechCraft https://www.arxiv.org/abs/2408.13608 , TextrolSpeech https://arxiv.org/abs/2308.14430) will require retraining to reproduce, but if possible, one of these should be compared against for emotion control.
2. There is no human evaluation in any of the experiments, which is the gold standard for evaluation; it would be good to have human speech-controllabel consistency metrics and human speech naturalness metrics.

**Questions:**

1. Did you try comparing your approach to learn disentangled style representations (MAE-based) to other approaches you mention in the related work section (e.g. NANSY, NaturalSpeech3, SpeechTokenizer)?
2. Is there any way to evaluate the disentanglement of style and content tokens apart from the swapping style and content tokens experiment you mention?
3. Lines 411-412: ‘We speculate that this may be due to larger scale values causing distortion in the generated speech, similar to the phenomenon observed with CFG in score-based models‘ : Could you provide a citation for this phenomenon for CFG in score-based models?

---

### Official Review · Reviewer_vuFP · 2024-11-04

**Soundness:** 3
**Presentation:** 3
**Contribution:** 3
**Rating:** 6
**Confidence:** 5

**Summary:**

- The authors have proposed a two-stage LM-based TTS approach. Two language models are leveraged in the proposed approach, with one (style LM) generating the style tokens and followed by another one (acoustic LM) generating the codec tokens.
- The style tokens consist of age, gender, pitch mean for average pitch, pitch std for the extent of pitch variation, emotion represented by arousal, dominance, and valence, snr for signal-noise rate, and C50 for reverberation level.
- CFG is used in the style LM.

**Strengths:**

- This paper focuses on fine-grained control in LM-based TTS systems, a topic that has garnered significant interest within the community.
- The inclusion of a demo webpage is an excellent addition.
- The idea should work well, due to the capability of in-context learning, if the style tokens are perfectly extracted.

**Weaknesses:**

- The review of fine-grained control in TTS systems is somewhat basic. The shortcomings of existing approaches and the rationale for using this particular style-LM in LM-based methods are not clearly addressed. To my knowledge, there are many relevant papers on this topic; please cite and discuss the pros and cons accordingly.

- The style-LM relies on external embedding or attribute extractors, whose quality, I believe, significantly impacts performance. Since perfect embeddings or attributes cannot always be extracted, it would be beneficial to analyze how these extractors affect the final output, especially given the issues with the arousal-valence-dominance extractor mentioned by the authors. Including an analysis on ground-truth audio would strengthen the paper’s conclusions.

- There is limited comparison between the proposed approach and existing models. For instance, it would be helpful to show the level of improvement over models like VALL-E (a fundamental LM-based approach) or NaturaSpeech3.

**Questions:**

In addition to addressing the identified weaknesses, it would be helpful for the authors to consider the following questions:

- I am trying to understand the advantage of using a 2-stage system over a 1-stage system with attributes. Based on Figure 5, significant benefits appear only in terms of the arousal-valence-dominance attributes. However, given the potential uncertainty associated with the wav2vec2 extractor for these attributes, this advantage may be limited. Could the authors elaborate on this?

- The selected attributes may not fully encompass all aspects of fine-grained control in TTS (for example, speaking speed is not included). Would it be possible to evaluate performance regression when introducing additional attributes during acoustic LM training? I would assume that adding more attributes could increase the training complexity. Insights on this would be valuable.

---

### Note · Authors · 2024-11-24

I have read and agree with the venue's withdrawal policy on behalf of myself and my co-authors.